# Directed self-assembly of viologen-based 2D semiconductors with intrinsic UV–SWIR photoresponse after photo/thermo activation

Xiao-Qing Yu[1,2], Cai Sun[1], Bin-Wen Liu[1], Ming-Sheng Wang ⬤ [1✉] & Guo-Cong Guo[1]

Extending photoresponse ranges of semiconductors to the entire ultraviolet–visible (UV)–shortwave near-infrared (SWIR) region (ca. 200–3000 nm) is highly desirable to reduce complexity and cost of photodetectors or to promote power conversion efficiency of solar cells. The observed up limit of photoresponse for organic-based semiconductors is about 1800 nm, far from covering the UV–SWIR region. Here we develop a cyanide-bridged layer-directed intercalation approach and obtain a series of two viologen-based 2D semiconductors with multispectral photoresponse. In these compounds, infinitely π-stacked redox-active N-methyl bipyridinium cations with near-planar structures are sandwiched by cyanide-bridged $Mn^{II}$–$Fe^{III}$ or $Zn^{II}$–$Fe^{III}$ layers. Radical–π interactions among the infinitely π-stacked N-methyl bipyridinium components favor the extension of absorption range. Both semiconductors show light/thermo-induced color change with the formation of stable radicals. They have intrinsic photocurrent response in the range of at least 355–2400 nm, which exceeds all reported values for known single-component organic-based semiconductors.

---

[1] State Key Laboratory of Structural Chemistry, Fujian Institute of Research on the Structure of Matter, Chinese Academy of Sciences (CAS), 155 Yangqiao Road West, 350002 Fuzhou, Fujian, China. [2] University of Chinese Academy of Sciences, No.19A Yuquan Road, 100049 Beijing, China. ✉email: mswang@fjirsm.ac.cn

 1

Photoresponse range is a metric that significantly affects the potentials of semiconductors for photodetection, solar energy conversion and other applications[1–4]. As for photodetection, detection ranges of commercial Si-based photodetectors and InGaAs photodetectors are mostly in the 200–1100 nm and 900–3000 nm regions, respectively[1,5]. To realize photodetection in the ultraviolet (UV)–shortwave infrared (SWIR) region (ca. 200–3000 nm), either military or civil photodetecting devices usually have to combine the above two photodetectors. This will increase cost and complexity of the desirable device. As for solar energy conversion, photoresponse band of the widely used commercial photovoltaic material silicon is about 300–1200 nm[6,7]. This range far from covers the whole solar spectrum (∼295–2500 nm), resulting in limited energy conversion fields of Si-based solar cells[2,8]. Constructing multiple blend systems or multi-junction device structures was demonstrated to be effect methods to fully utilize the solar radiation, but complexity, manufacture cost, and sometimes stability of the device should be concerned before stepping into the market[9–11]. The above issues for both photodetection and solar energy conversion devices are promisingly avoided, if a single semiconductor material with a photoresponsive range covering the entire UV–SWIR region is applied[10,12]. As reported, some single-component inorganic photoelectronic semiconductors showed strikingly wide detection ranges, for example, the photoresponse range of SnTe covers 254–4650 nm[13]. However, practical application of them is still greatly limited by material rigidity, as well as complex and expensive manufacturing processes[14,15]. Organic-based photoelectronic semiconductors are characteristic of flexibility and facile preparation, but the observed up limit for photoresponse is about 1800 nm[10,12,15–18]. It is still of importance to develop effective and general design methods for single-component organic-based semiconductors with intrinsic photoresponse in the entire UV–SWIR range.

Viologen (N,N′-disubstituted bipyridinium) compounds are good candidates for single-component organic-based semiconductors with broadband photoresponse. Firstly, strong cation···π interactions between viologen components favor the construction of organic semiconductors, and conductance and photoconductance of viologen-based semiconductors may dramatically increase after photoinduced electron transfer (PET) and generation of free radical products[19–21]. Secondly, single viologen cation usually has a red-shifted absorption band after forming a radical species. When viologen radicals are further closely π-stacked, radical–π interactions that are stronger than cation–π, and π–π interactions will make energy gap narrower and correspondingly absorption band much broader[22–24]. Even so, improving the following two properties is still needed for viologen compounds. Firstly. photoconversion rate in bulk media is usually low, owing to low penetrability of UV–Visible light[25,26]. Thermal treatment may avoid this problem, because heat easily conducts to the whole media. Many examples have indicated viologen

compounds are probable to undergo heat-induced electron transfer (HET) when they tend to be planar[27,28]. Therefore, thermo-active viologen semiconductors are prospectively constructed with near-planar/planar viologen cations. Secondly, the radical products are highly active and easily faded in an oxygen atmosphere with the presence of heat or not[29,30]. Several examples have demonstrated that, the stability of viologen radicals can be clearly improved when the radicals are loaded in a porous framework[31,32] or sandwiched by inorganic layers[33] to avoid contact with oxygen.

In this work, we present a cyanide-bridged layer-directed intercalation approach to realize all above points and obtain single-component viologen-based semiconductors with intrinsic UV–SWIR photoresponse ability, photo/thermo activeness, and very long lifetime of radical products. Each cyanide-bridged layer in Prussian blue or its analogs with perovskite-like structures has periodically arranged hexacoordinated metal sites and limited metal-to-metal distances (Fig. 1)[34]. The periodic arrangement of metal coordination sites facilitates the orderly and infinite accumulation of axial ligands. The closest and non-contact metal-to-metal distances are usually 7.6 Å[34], which are suitable to support two π stacking interactions[35]. As shown in Fig. 1, a sandwiched inorganic–organic hybrid structure with infinitely stacked organic supramolecular layers will be formed, when metals with these distances are all coordinated by one viologen ligand or its analogs and then the layers are intercalated. Coexistence of π···π and cation···π interactions in the organic supramolecular layer is predicable since viologen and its analogs are aromatic cations. As mentioned above, this case favors the construction of a semiconductor. In addition, the viologen ligand or its analogs will become planarization owing to the close stacking of adjacent ligands, which may bring thermo activeness as stated above. Based on these considerations, we integrated the redox photoactive N-methyl bipyridinium (MQ$^+$) cation into cyanide-bridged M$^{II}$–Fe$^{III}$ (M$^{II}$=Mn$^{II}$ or Zn$^{II}$) layers as axial ligands, and obtained a series of two viologen-based 2D semiconductors, [{M$^{II}$(MQ)$_2$}{Fe$^{III}$(CN)$_6$}]Cl·3H$_2$O (**1**, M=Mn; **2**, M=Zn). These semiconductors are both thermo and light active in the crystalline state. After PET or HET, they generate long-lived radical products, and show intrinsic photoresponsive bands covering the UV–SWIR region (at least 355–2400 nm, monitored using our lasers).

## Results

**Directed synthesis and structural characterization.** Single crystals of **1** and **2** were all obtained from the diffuse reaction of MnCl$_2$·5H$_2$O, MQCl·H$_2$O, and K$_3$[Fe$^{III}$(CN)$_6$] in a molar ratio of 1:2:1 in water. Powder X-ray diffraction (PXRD; Supplementary Fig. 1) and elemental analysis demonstrated phase purity of the obtained crystalline samples. Only the crystal structure of **1** is described here since **1** and **2** are isostructural. As can be seen from Fig. 2a, cyanide-bridged layers of **1** are intercalated through

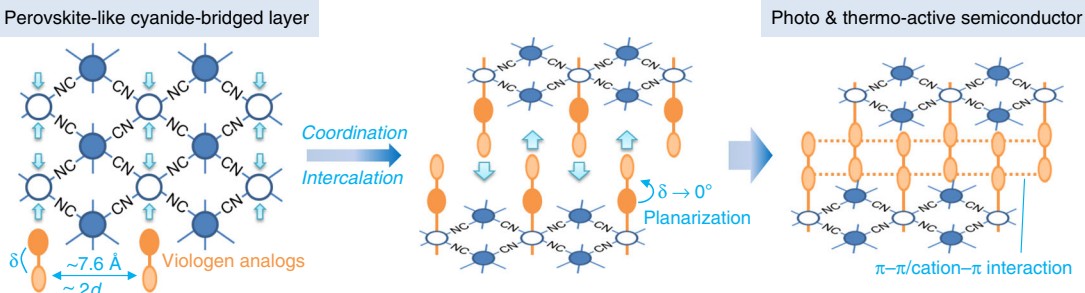

**Fig. 1 Design strategy in this work.** δ and $d_{\pi\text{-}\pi}$ denote the interannular angle and common separation for π–π interactions, respectively.

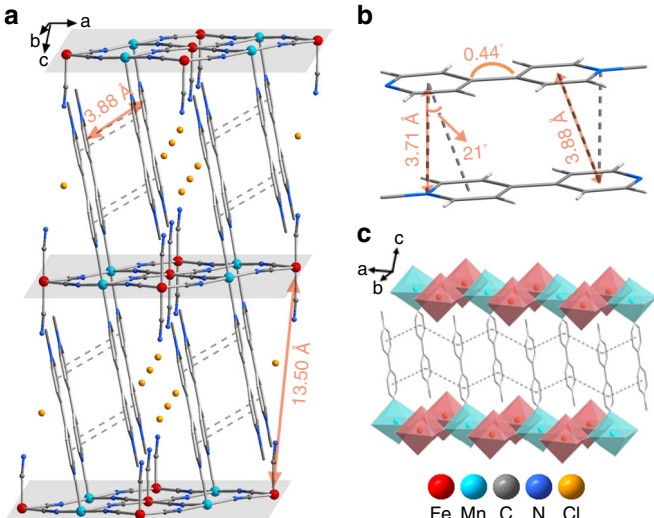

**Fig. 2 Crystal structure of 1. a** Side view of the 3-D packing structure; **b** two π-stacked MQ$^+$ cations; **c** infinitely π-stacked MQ$^+$ cations between two perovskite-like cyanide-bridged layers (cyano groups are drawn as vertexes of octahedra). Dash lines depict π stacking interactions.

π⋯π and cation⋯π interactions between two adjacent MQ$^+$ ligands. Each Fe$^{III}$ atom coordinates to six cyano groups (four, bridged, two, mono-coordinated), while each Mn$^{II}$ atom is ligated by two MQ$^+$ ligands and four bridged cyano groups from four [Fe(CN)$_6$]$^{3-}$ units. The interannular angle of each MQ$^+$ cation is 0.44°, close to be planar (Fig. 2b). Centroid (pyridyl)⋯centroid (pyridyl) and N⋯centroid (pyridyl) distances between two adjacent π-stacked MQ$^+$ cations are 3.88 and 3.71 Å, respectively (Fig. 2b). Every π-stacked MQ$^+$ layer is sandwiched by two cyanide-bridged layers, which offers a chance to shield the air (Fig. 2c). After standing in the dark under the 100% relative humidity for 12 h, the samples still kept the same PXRD patterns to the as-prepared crystalline samples (Supplementary Fig. 1), illustrating high wet stability for both compounds. Thermo-gravimetric (TG) analyses indicated that both compounds contain water molecules (Supplementary Fig. 2). In the electron absorption spectrum (Fig. 3), a series of infrared (IR) overtone peaks for H$_2$O can be observed, such as at 1412, 1916, and 2158 cm$^{-1}$. Crystal structures of both compounds can be retained after thermal annealing at 180 °C and removing all the water molecules, as evidenced by the similar PXRD patterns between as-prepared and thermally annealed samples (Supplementary Fig. 1). Therefore, the following mentioned thermal annealing processes were all operated at 180 °C.

**Photo/thermo-induced coloration.** Both **1** (Fig. 3) and **2** (Supplementary Fig. 3) may undergo photo/thermo-induced coloration. Also, only compound **1** is described in detail. The as-prepared crystalline sample (**1A**) of **1** changed its color from brown to black upon irradiation by a Xe lamp (~200 mW cm$^{-2}$) at room temperature (Fig. 3a). The photoinduced black sample (**1B-P**) appeared a broad electron absorption band around 620 nm and a much broader band in the range of ~900–2500 nm (Fig. 3a). There was no further clear variation of the electron absorption spectrum when the sample was irradiated beyond 100 min. Upon thermal annealing under 180 °C in air, the **1A** sample also turned black (Fig. 3b). The yielded black sample (**1B-T**) generated similar but stronger electron absorption bands (Fig. 3b) and electron spin resonance (ESR) signals (Fig. 4b) to those of **1B-P**. If UV/Visible/NIR spectra and IR spectra are combined, we can see that the absorption band of **1B-T** extends

to 3000 nm (Fig. 4a and Supplementary Fig. 4), that is to say, the absorption spectrum of **1B-T** covers the whole UV–SWIR region. The absorption spectrum did not change again after annealing for about 150 min (Fig. 3b). **1B-T** was considerably stable because its absorption spectrum almost retained after standing in the dark in air at room temperature for six months (Fig. 4a). Similar to **1B-T**, **2B-T** also shows broad absorption in the UV–SWIR region and high stability in air (Supplementary Fig. 5a).

**Electrical studies.** Electron absorption (Fig. 3 and Supplementary Fig. 3) and ESR (Fig. 4b, Supplementary Fig. 5b, Supplementary Note 1, and Supplementary Table 2) data of **1** and **2** revealed that thermal annealing triggered higher conversion rate than the irradiation method. So, both thermo-induced samples **1B-T** and **2B-T** were selected to perform electric tests. A well-known two-probe method using silver paste for a pellet sample was adopted[36]. The current–voltage (I–V) characteristic curves before and after coloration for **1B-T** showed a symmetrical nearly linear relationship at room temperature, which indicated that the sample formed an Ohmic contact and the carriers derived from intrinsic thermal excitation. After HET, the conductivity increased ~4-folds (Supplementary Fig. 6), which is accordance with the decrease of activation energy (Supplementary Fig. 7). This phenomenon is consistent with the well-established conclusion that receiving electrons and forming a radical species is beneficial to improve the conductivity of one π aggregate[18,20]. As mentioned above, the **1B-T** sample had an intrinsic absorption spectrum covering the whole UV–SWIR region (Supplementary Fig. 4). Owing to limited testing equipment, we selected one 355 nm diode pumped solid-state laser and the other OPO laser with tunable wavelength ranging from 410 to 2400 nm to monitor the photocurrent response in the UV and visible–SWIR regions, respectively. Photocurrent gain can be expressed by the relative magnitude of the current change, $(I_{irr} - I_{dark})/I_{dark}$. Figure 5 illustrates that the **1B-T** sample had photocurrent response in the range of 355–2400 nm. The other two samples showed the similar photocurrent curves and photocurrent gains, indicating that the photoresponsive behavior of **1B-T** is repeatable (Supplementary Fig. 8). Meanwhile, the greater the illumination power, the greater the photocurrent gain under laser irradiation (Supplementary Fig. 9). Bulk electric tests of **2B-T** indicated that **2B-T** was also an intrinsic semiconductor and showed photocurrent response in the range of 355–2400 nm (Supplementary Figs. 10 and 11). The photoresponse ranges for both **1B-T** and **2B-T** exceed all known reported values for single-component organic-based semiconductors[5,10,12,37,38].

**Discussion**

The cyanide-bridged inorganic layers in **1** are similar to those of the room-temperature phase of RbMn$^{II}$[Fe$^{III}$(CN)$_6$]·H$_2$O. The latter has a CN → Fe$^{III}$ charge-transfer band at ~410 nm (3.02 eV), a d–d transition band of Fe$^{III}$ at ~520 nm (2.38 eV), and a MMCT band at ~680 nm (1.82 eV)[39]. As illustrated in Fig. 3, the **1A** sample also contains these electron-transition bands. The photoresponsive range of **1A** is in ~280–420 nm, wherein the optimal wavelength is around 320 nm (3.88 eV). Calculations of band structure and partial density of states showed that the absorption around 320 nm can be assigned to Cl$^-$/CN$^-$/Fe$^{III}$ → MQ$^+$ electron transitions (Supplementary Fig. 12). The newly emerged bands around 620 nm (2.00 eV) for **1B-P** and **1B-T** are characteristic of single MQ$^•$ radical[40], indicating that the MQ$^+$ ligand received an electron upon irradiation or thermal annealing. ESR signal of the MQ$^•$ radical was not clear for **1B-P**, but was identifiable for **1B-T** (at $g = 2.005$; Fig. 4b). Low penetration of light in the crystal resulted in the generation

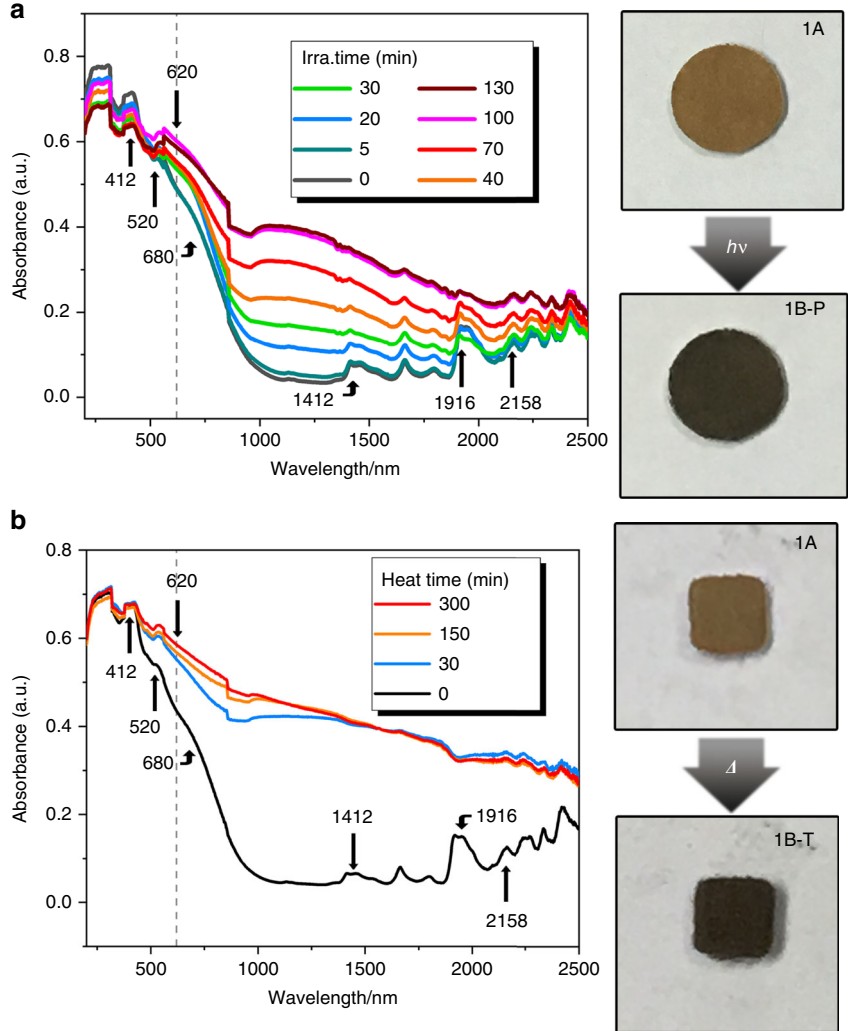

**Fig. 3 Electron absorption spectra and color change.** Time-dependent electron absorption spectra of **1** measured in the diffuse reflectance mode and color change upon irradiation (**a**) or thermal annealing at 180 °C (**b**).

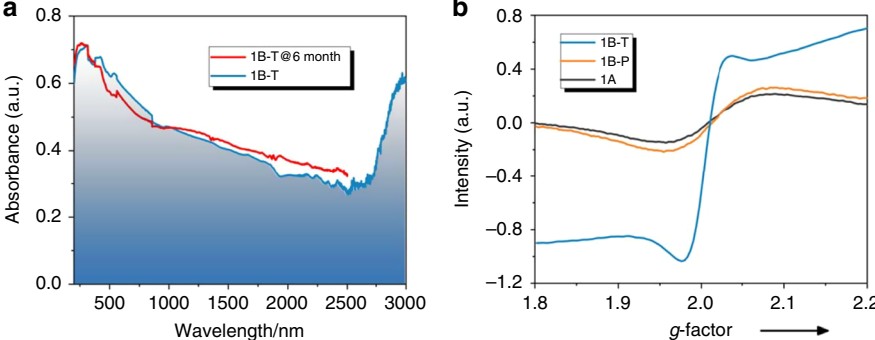

**Fig. 4 Air-stability and ESR test. a** Stability test for the thermo-induced sample **1B-T** monitored through the combination of UV/Visible/NIR spectra (200–2500 nm) and IR spectra (2500–3000 nm). The sample was placed in air in the dark at room temperature for 6 months. The minor difference between **1B-T** and **1B-T@6month** was attributed to systematic errors caused by instrumental instability or shift of the sample holder. **b** ESR patterns for **1A**, **1B-P**, and **1B-T**. Note: **1A** is the as-prepared crystalline sample; **1B-P** is the sample that was irradiated by an Xe lamp for 70 min; **1B-T** is the sample that was thermally annealed at 180 °C for 150 min.

of a small number of radicals, whose ESR signals were covered by Mn$^{II}$ signals. In comparison, the heating mode led to the generation of more radicals. As for **2**, ESR signals for both photo/thermo-induced samples were clear without the shielding of Mn$^{II}$ signals (Supplementary Fig. 5b).

All photo/thermo-induced samples of **1** showed PXRD patterns similar to that of the as-prepared sample (Supplementary Fig. 1). Therefore, the framework of **1** did not undergo large isomerization or decomposition during its coloration. The lattice water molecules in **1** can be excluded to be an electron donor for

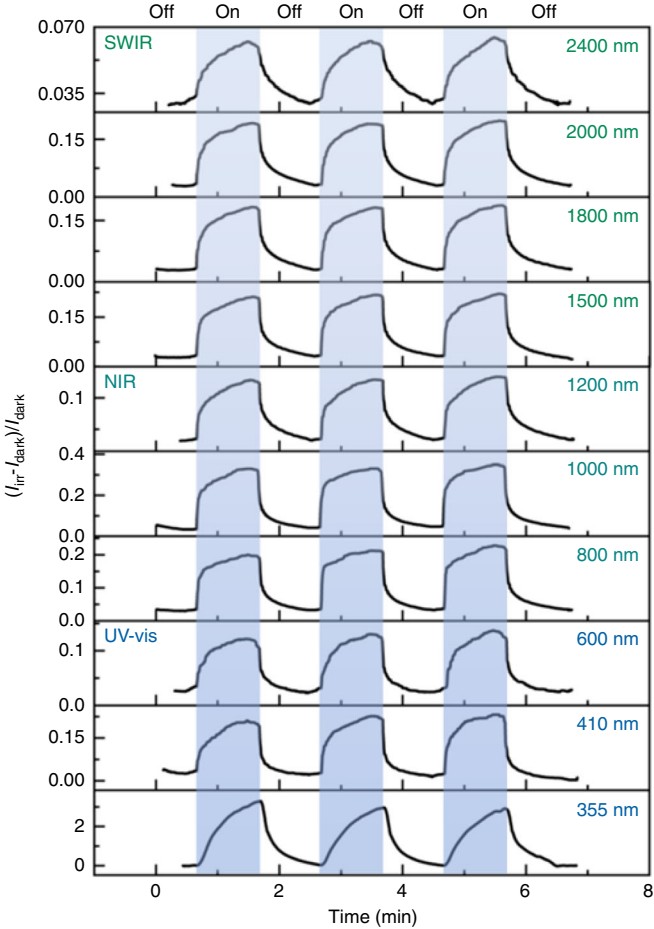

**Fig. 5 Photoresponse behavior of 1B-T monitored using lasers with different wavelengths.** A Newport Co. Pulseo GKNQL-355-3-30 diode pumped solid-state laser (70 kHz, 39 ns pulse width) and an OPO laser (10 Hz, 10 ns pulse width) are used for 355 nm and 410–2400 nm light, respectively. Powers for each wavelength: 355 nm, 2.1 W; 410 nm, 7.5 mW; 600 nm, 5.0 mW; 800 nm, 15 mW; 1000 nm, 25 mW; 1500 nm, 23 mW; 1800 nm, 22 mW; 2000 nm, 25 mW; 2400 nm, 6.0 mW.

MQ$^+$, because **1B-T** contained no water and also displayed electron transfer (Supplementary Fig. 2 and Fig. 3). In the crystal structure of **1** (Supplementary Fig. 13), each free Cl$^-$ ion forms four C(MQ$^+$)–H···Cl hydrogen bonds (H···Cl, 2.67 Å) with adjacent four MQ$^+$ ligands, and each mono-coordinated CN$^-$ group connects to two MQ$^+$ ligands through C(MQ$^+$)–H···N(CN) hydrogen bonds (H···N, 2.48 Å). According to the Marcus electron transfer theory[41], the shorter the separation between an electron donor and an electron acceptor is, the faster the electron transfer process. It seems that the electron transfer rate through the C(MQ$^+$)–H···N(CN) path should be faster than the C(MQ$^+$)–H···Cl path. However, as evidenced by the references, both Cl$^-$[32,42–44] and CN$^-$[45,46] ions may act as effective electron donors for viologen cations. X-ray photoelectron spectroscopy is a common tool to prove the direction of electron transfer, but the paramagnetic metals in **1** are easily reduced by X-ray-induced electrons, which brings the difficulty for analysis. Searching other applicable methods is still underway.

The multispectral photocurrent response of **1B-T** and **2B-T** originates from the ultrabroad electron absorption (Supplementary Figs. 3 and 4). Time-dependent density functional theoretical (TD-DFT) calculations (Supplementary Fig. 14)[47], using simplified structural models truncated form the crystal structure, indicates that the emerged broad bands in the range of

~900–3000 nm for all photo/thermo-induced samples can be assigned to the contribution of π-stacked MQ aggregates, which in the form of [(HMQ$^{2+}$)(HMQ$^{+•}$)]$_n$ or [(HMQ$^+$)(HMQ$^{+•}$)]$_n$ ($n$ refers to the number). In summary, aiming to achieve single-component semiconductors with intrinsic UV–SWIR photo-response, a cyanide-bridged layer-directed intercalation approach has been firstly demonstrated. The obtained two viologen-based 2D semiconductors have intrinsic absorption bands covering the whole UV–SWIR region after photo/thermo activation. They show photocurrent response at least in the wavelength range of 355–2400 nm (monitored using our limited lasers), which makes a record for single-component organic-based semiconductors. The design strategy explored in this work may inspire the synthesis of new organic-based semiconductors for broadband photodetectors and solar cells. In the following work, we will further improve electrical properties (conductivity, photocurrent gain, etc.) and photoresponse bands (extending to the mid-IR region), and explore effective methods to construct optical devices.

## Methods

**Materials.** MnCl$_2$·5H$_2$O, ZnCl$_2$ and K$_3$[Fe(CN)$_6$] in AR grade were purchased commercially. They were directly used without further purification. Water was deionized and distilled before use. MQCl·H$_2$O (MQ$^+$=$N$-methyl-4,4′-bipyridinium) was synthesized according to the same procedure reported in the literature[48].

**Syntheses of [{Mn$^{II}$(MQ)$_2$}{Fe$^{III}$(CN)$_6$}]Cl·3H$_2$O (1).** Typically, a 50 mL small beaker was placed in a 300 mL big one, which was filled with distilled water to approximately 0.5 cm above the top of the small beaker. A frozen 2 mL aqueous solution of MQCl·H$_2$O (899 mg, 4 mmol) and MnCl$_2$·5H$_2$O (432 mg, 2 mmol) was thrown into the bottom of the small beaker, while the other frozen 2 mL aqueous solution of K$_3$[Fe(CN)$_6$] (659 mg, 2 mmol) was put into the bottom of the big beaker. The big beaker was sealed with a plastic wrap and allowed to stand in the dark at room temperature for one week to yield dark brown cubic crystals. The crystals were filtered, washed with water and ethanol, and finally dried in air for 1 day. Yield based on K$_3$[Fe(CN)$_6$]: 40% for **1**. All crystal samples for tests were carefully selected under microscope. The phase purity of all as-synthesized crystalline samples was checked via PXRD (Supplementary Fig. 1) and elemental analyses. Anal. Calcd (%) for C$_{28}$H$_{28}$ClFeMnN$_{10}$O$_3$: C, 48.12; H, 4.04; N, 20.04; Fe, 7.99; Mn, 7.86. Found: C, 48.14; H, 3.78; N, 20.57; Fe, 7.61; Mn, 7.26.

**Syntheses of [{Zn$^{II}$(MQ)$_2$}{Fe$^{III}$(CN)$_6$}]Cl·3H$_2$O (2).** Typically, a 50 mL small beaker was placed in a 300 mL big one, which was filled with distilled water to approximately 0.5 cm above the top of the small beaker. A frozen 2 mL aqueous solution of MQCl·H$_2$O (899 mg, 4 mmol) and ZnCl$_2$ (273 mg, 2 mmol) was thrown into the bottom of the small beaker, while the other frozen 2 mL aqueous solution of K$_3$[Fe(CN)$_6$] (659 mg, 2 mmol) was put into the bottom of the big beaker. The big beaker was sealed with a plastic wrap and allowed to stand in the dark at room temperature for one week to yield yellow plate crystals. The crystals were filtered, washed with water and ethanol, and finally dried in air for 1 day. Yield based on K$_3$[Fe(CN)$_6$]: 25% for **2**. All crystal samples for tests were carefully selected under microscope. The phase purity of all as-synthesized crystalline samples was checked via PXRD (Supplementary Fig. 1) and elemental analyses. Anal. Calcd (%) for C$_{28}$H$_{28}$ClFeZnN$_{10}$O$_3$: C, 47.42; H, 3.98; N, 19.75; Fe, 7.87; Zn, 9.22. Found: C, 45.54; H, 3.38; N, 19.75; Fe, 7.43; Zn, 10.18.

**Measurements.** IR spectra were recorded on a PerkinElmer Spectrum One FT-IR spectrometer using KBr pellets in the range of 4000–450 cm$^{-1}$. Thermogravimetric analysis was conducted on a Mettler TOLEDO simultaneous TGA/DSC apparatus. Elemental analyses of C, H and N were measured on an Elementar Vario EL III microanalyzer, while those of Fe, Zn, and Mn were measured on an ULTIMA 2 ICP Optical Emission Spectrometer. Electron spin resonance (ESR) spectra were recorded on a Bruker ER-420 spectrometer with a 100 kHz magnetic field in the X band at room temperature. Powder X-ray diffraction (PXRD) patterns were collected on a Rigaku Desktop MiniFlexII diffractometer using Cu $K_α$ radiation ($λ$ = 1.54056 Å) powered at 30 kV and 15 mA. Diffuse reflectance spectra were recorded at room temperature in the wavelength range of 200–2600 nm on a PerkinElmer Lambda 900 UV/vis/NIR spectrophotometer equipped with an integrating sphere. A BaSO$_4$ plate was used as the reference (100% reflection), on which the finely ground powder of the sample was coated. Photoirradiation for coloration was carried out with a PLS-SXE300D 50-W xenon lamp system, wherein an IR filter was applied. $I$–$V$ curves were measured in a Keithley 4200-SCS semiconductor parameter analyzer using pellet samples by the two-probe method using silver paste. An OPOTEK Vibrant laser (10 Hz; 10 ns pulse width; spot size, ca. 1–2 cmφ)

and a Newport Co. Pulseo GKNQL-355-3-30 diode pumped solid state (DPSS) laser (355 nm; 70 kHz; 39 ns pulse width; spot size, ca. 1.5 cmφ) were used for photocurrent tests.

**Single-crystal X-ray crystallographic study.** Single-crystal X-ray diffraction measurements of **1** and **2** were performed on a Rigaku SATURN70 CCD diffractometer, using graphite monochromated Mo $K_{\alpha}$ radiation ($\lambda = 0.71073$ Å). Intensity data sets were collected using scan techniques, and corrected for $Lp$ effects. The primitive structures were solved by the direct method using the Siemens SHELXTL$^{TM}$ Version 5 package of crystallographic software[49]. Difference Fourier maps based on these atomic positions yielded other non-hydrogen atoms. The final structures were refined using a full-matrix least-squares refinement on $F^2$. All non-hydrogen atoms were refined anisotropically. H atoms on $N$-substituted C atoms were not included for their symmetrical disorder, and those of lattice water molecules were also not added for weak diffraction. Other H atoms were generated geometrically. Supplementary Table 1 shows the crystal and structure refinement data.

**Calculations of electron absorption spectra.** All calculations were performed with the time-dependent density functional theory (TD-DFT) method at the pbe1pbe/6-31 g* level using the Gaussian 09 software package[50]. Calculation models were truncated from the crystal structure of **1** and modified with H atoms replacing metal atoms.

**Calculations of band structures and partial density of states.** All calculations were executed using the Cambridge Sequential Total Energy Package (Castep)[51]. A plane-wave energy of 400 eV and a $3 \times 3 \times 2$ Monkhorst-Pack grid of $k$-points were selected. The exchange-correlation energy was described by the Perdew–Burke–Eruzerhof (PBE) functional within the generalized gradient approximation (GGA)[52]. The norm-conserving pseudopotentials were chosen to modulate the electron–ion interaction[53]. Pseudo atomic calculations were performed for C $2s^2 2p^2$, H $1s^1$, N $2s^2 2p^3$, O $2s^2 2p^4$, Cl $3s^2 3p^5$, Fe $3d^6 4s^2$, and Mn $3d^5 4s^2$. Other parameters used in the calculations were set by the default values of the CASTEP code.

## Data availability

The X-ray crystallographic data (**1** and **2**) reported in this study have been deposited at the Cambridge Crystallographic Data Center (CCDC), under deposition number CCDC 1958300–1958301. These data can be obtained free of charge from The Cambridge Crystallographic Data Center via http://www.ccdc.cam.ac.uk/conts/retrieving.html or from the Cambridge Crystallographic Data Center, 12 Union Road, Cambridge CB2 1EZ, U.K. Fax: (Internet) +44-1223/336-033. E-mail: deposit@ccdc.cam.ac.uk. We declare that the main data supporting the findings of this study are available within the article and its Supplementary Information files. All relevant source data are also available from the corresponding author upon reasonable request.

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

## Acknowledgements

This work was supported by the National Natural Science Foundation of China (91545201, 21827813), Key Research Program of Frontier Science, CAS (QYZDB-SSW-SLH020, QYZDJ-SSW-SLH028), and the Youth Innovation Promotion Association of the Chinese Academy of Sciences.

## Author contributions

M.S.W. conceived the idea and designed the experiments. X.Q.Y. did the synthesis, TGA, FT-IR, UV–Vis, PXRD, ESR, elemental analysis, electrical measurements, solved the crystal structures and performed the DFT calculations. C.S. analyzed the data. B.W.L. assisted X.Q.Y. in testing photocurrent measurements. M.S.W. and G.C.G. wrote the manuscript.

## Competing interests

The authors declare no competing interests.
