## [Peer Review File · Nature Communications]

Reviewers' comments:

Reviewer #2 (Remarks to the Author):

Broadband materials are highly desirable for many fields, such as photoelectronic detection, solar energy conversion, photocatalysis and military camouflage. As far as I know, materials with photocurrent response in the whole UV-SWIR range are scarce, and no examples have been documented for organic-based materials. Viologens and their analogues are well known for their remarkable absorption in the visible region after receiving electrons and forming radicals. Broader absorption is predictable if they are consolidated through infinite π -stacking, and this point is useful to design the UV-SWIR photoresponsive materials. However, it is indeed very difficult to realize infinite π -stacking by design in the crystal engineering field. Besides, improving the conversion rate of radicals and stability of the highly active radicals are clearly of significance for applications of all electron-transfer photo/thermo/electro/piezochromic compounds. In this paper, Wang and coworkers present a cyanide-bridged layer-directed intercalation approach that effectively addresses the above issues. By this approach, a series of two photo & heat-active viologen-based 2D semiconductors with intrinsic absorption bands covering the entire UV-SWIR range were obtained. They showed photocurrent response in the wavelength range of at least 355–2400 nm, which exceeds all reported values for known single-component organic-based semiconductors. The approach described in this manuscript will not only promote the development of crystal engineering and photo/thermo/electro/piezochromic fields, but help the design of broadband materials for photodetection, solar energy conversion and other fields. I strongly recommend publication of this work in Nature Communications after considering the following points.

1. Generally, perovskite materials have poor stability. They are easily decomposed in a humid or high temperature environment, resulting in serious degradation of the device. This series of perovskite-like materials can exist in a high temperature environment, but are they stable in the humid environment?

2. Figure S11 shows photoresponse behavior of the 1B-T to 1000 nm laser at different powers. Whether other wavelengths have a similar linear correlation between relative photocurrent gain and power?

3. The sentence "The current-voltage (I-V) characteristic curves before and after coloration for 1B-T showed a symmetrical linear relationship at room temperature, which indicated that the sample formed an Ohmic contact and the carriers derived from intrinsic thermal excitation" indicates that the carriers of 1B-T come from intrinsic thermal excitation. It would be better to provide activation energy for 1A and 1B.

Reviewer #3 (Remarks to the Author):

This manuscript reported the designed syntheses of two viologen-based 2D semiconductors with UV-SWIR photoresponse by an interesting cyanide-bridged layer-directed intercalation approach. Material characteristics as well as the relation between photoresponsive characters and structures have been well studied. Besides interesting wide-range photoresponse exhibiting in the designed materials, the proposed approach presenting here gives a way to controllably assemble infinite π -stacking moieties, realize thermo-activation, and enhance the stability of radicals. As organic-based materials, such as perovskites having limited photoresponsive bands usually below 1800 nm, this work shows an efficient strategy to achieve materials with UV-SWIR photoresponsive bands, which reduces complexity of photoelectric devices and improves photo-energy efficiency. It contributes to the designed syntheses of new organic-based semiconductors for broadband photodetectors, solar cells and photocatalysts. Overall, this paper is well organized and written, which is really interesting for the publication in nature communications. Points are concerned by the Referee listed as follows:

1. Regarding to the electric test, pellet samples were used. How about single crystals?
2. As can be seen from Figure S7, ESR signals of the thermo-induced sample are stronger than those of the photo-induced sample. Comparison of relative radical concentration is needed.
3. Figure S11d shows only the relative relationship between photocurrent and powder under 1000 nm laser irradiation. More data using other wavelengths should be added.
4. In the titles of Figure S1 and Figure S7, the labels of the compound need to be corrected. For instance, 1B-T/ 1B-P should replace 2A-T/ 2A-P.

2. Reviewer 2

Comment 1: Generally, perovskite materials have poor stability. They are easily decomposed in a humid or high temperature environment, resulting in serious degradation of the device. This series of perovskite-like materials can exist in a high temperature environment, but are they stable in the humid environment?

Response 1: Yes, they are stable in the humid environment. As shown in the following figure, after standing in the dark under the 100% relative humidity for 12 h, the samples of compounds **1** and **2** (named as *1A_100% hum* and *2A_100% hum*, respectively) exhibited the same powder X-ray diffraction patterns (PXRD) to the as-synthesized samples. This means that the structures of **1** and **2** were still retained in the high-humidity environment.

The related discussion has been added to **the first paragraph of page 3 in the manuscript** as “*After standing in the dark under the 100% relative humidity for 12 h, the samples still showed the same PXRD patterns to the as-prepared crystalline samples (Figure S1), illustrating high wet stability for both compounds.*” Meanwhile, **Figure S1** in the old

Supporting Information has been updated with the newly recorded data *1A_100% hum* and *2A_100% hum*.

Revised **Figure S1** in the Supporting Information. PXRD patterns: a) **1_simulated**, **1A**, **1B-P**, **1B-T**, **1A_100% hum**; b) **2_simulated**, **2A**, **2B-P**, **2B-T**, **2A_100% hum**. Note: **1_simulated/2_simulated** refer to simulated PXRD curves that were simulated using the single-crystal X-ray diffraction data. **1A/2A** are as-prepared crystalline samples; **1B-P/2B-P** are samples that were irradiated by an Xe lamp for 70 min; **1B-T/2B-T** are samples that were thermally annealed at 180 °C for 150 min; **1A_100% hum/2A_100% hum** are samples that were placed in the dark under the 100% relative humidity for 12 h.

Comment 2: **Figure S11** shows photoresponse behavior of the **1B-T** to 1000 nm laser at different powers. Whether other wavelengths have a similar linear correlation between relative photocurrent gain and power?

Response 2: Yes, as shown in the following figure, **1B-T** has a similar linear correlation between relative photocurrent gain and power at other wavelengths, such as 1200 nm and 1500 nm. The linear correlation between relative photocurrent gain and power means that a higher power density produces more carriers in a specific optical power range.

Relevant experimental data have been added to the revised Supporting Information as **Figure S12**. Meanwhile, the original description “*the greater the illumination power, the greater the photocurrent gain under 1000 nm laser irradiation (Figure S11d)*” in the **first paragraph of page 6 in the manuscript** has been changed to “*the greater the illumination power, the greater the photocurrent gain under laser irradiation (Figure S12).*”

Revised **Figure S12** in the Supporting Information. Laser power-dependent photoresponse behavior of **1B-T** with 1000 nm, 1200 nm and 1500 nm lasers, respectively. Inset: Plot of relative photocurrent gain versus power. As can be seen, photocurrent gain shows a linear correlation to power, which means that a higher power density produces more carriers in a specific optical power range.

Comment 3: The sentence “*The current–voltage (I – V) characteristic curves before and after coloration for **1B-T** showed a symmetrical linear relationship at room temperature, which indicated that the sample formed an Ohmic contact and the carriers derived from intrinsic thermal excitation*” indicates that the carriers of **1B-T** come from intrinsic thermal excitation. It would be better to provide activation energy for **1A** and **1B**.

Response 3: According to your recommendation, we have now measured *in situ* temperature-dependent conductivities of **1** in air using the same pellet to achieve activation energies E_a of **1A** and **1B-T**. As shown in the following figure, the activation energy E_a decreased after thermal annealing, which is in accordance with the increasing of conductivity.

Relevant experimental data have been added to the revised Supporting Information as **Figure S10**. Meanwhile, the original description “*After HET, the conductivity increased ~4 folds (Figure S10).*” in **the first paragraph of page 6 in the manuscript** has been changed to “*After HET, the conductivity increased ~4 folds (Figure S9), which is accordance with the decrease of activation energy (Figure S10).*”

Revised **Figure S10**. Temperature-dependent conductivities of **1A** and **1B-T**. $\ln\sigma = -E_a/k_B T + \text{constant}$, where E_a is the activation energy, k_B is Boltzmann constant, T is temperature, σ is conductivity.

2. Reviewer 3

Comment 1: Regarding to the electric test, pellet samples were used. How about single crystals?

Response 1: We have tried to test electrical properties with single crystals. However, in this series of compounds, it is difficult to ensure that the direction of electrode is the same as that of π -stacking on account of the irregular shape of the single crystal. To ensure the accuracy and repeatability of the compound performance, we performed the electric test with pellet samples.

Comment 2: As can be seen from Figure S7, ESR signals of the thermo-induced sample are stronger than those of the photo-induced sample. Comparison of relative radical concentration is needed.

Response 2: According to your recommendation, we have measured the spin densities of equivalent **1** and **2** using a standard sample set in the instrument as a reference. The test conditions and methods are consistent with the standard sample. The following table shows the relative spin densities that are obtained by subtracting the recorded data to that of **1A** or **2A**. **1B-P/2B-P** are samples that were irradiated by an Xe lamp for 70 min; **1B-T/2B-T** are samples that were thermally annealed at 180 °C for 150 min.

Relevant experimental data have been added to the revised Supporting Information as **Table S2**. Meanwhile, the original description “*Electron absorption (Figure 3 and S3) and ESR (Figure S7) data of 1 and 2 revealed that thermal annealing triggered higher conversion rate than the irradiation method.*” in the first paragraph of page 6 in the manuscript has been changed to “*Electron absorption (Figure 3 and S3) and ESR (Figure 4b, S5b and Table S2) data of 1 and 2 revealed that thermal annealing triggered higher conversion rate than the irradiation method.*”

Table S2 in the revised supporting information. Relative spin densities.

Sample	Relative spin density / mol ⁻¹	Sample	Relative spin density / mol ⁻¹
1B-P	2.478×10^{21}	2B-P	1.020×10^{17}
1B-T	5.637×10^{22}	2B-T	1.303×10^{20}

Comment 3: Figure S11d shows only the relative relationship between photocurrent and powder under 1000 nm laser irradiation. More data using other wavelengths should be added.

Response 3: As described in **Comment 2 of Reviewer 2**, we have additionally monitored photoresponse behavior of **1B-T** at different powers with 1200 nm and 1500 nm laser. The experimental result showed that **1B-T** has also a similar linear correlation between relative photocurrent gain and power at 1200 nm and 1500 nm. Please see more discussion in **Comment 2 of Reviewer 2**.

Comment 4: In the titles of Figure S1 and Figure S7, the labels of the compound need to be corrected. For instance, **1B-T/ 1B-P** should replace **2A-T/ 2A-P**.

Response 4: Thanks, these are typing errors. We have now made corresponding modifications. The original title of **Figure S1** “*PXRD patterns: a) 1A, 1B-P, 1B-T; b) 2A, 2B-P, 2B-T, 2C. The simulated PXRD curve is based on the single-crystal X-ray diffraction data. 1A/2A are as-prepared crystalline samples; 2A-P/2B-P are samples that were irradiated by an Xe lamp for 70 min; 2A-T/2B-T are samples that were thermally annealed at 180 °C for 150 min.*” has been changed to “*PXRD patterns: a) 1_simulated, 1A, 1B-P, 1B-T, 1A_100% hum; b) 2_simulated, 2A, 2B-P, 2B-T, 2A_100% hum. Note: 1_simulated and 2_simulated refer to simulated PXRD curves that were simulated using the single-crystal X-ray diffraction data. 1A/2A are as-prepared crystalline samples; 1B-P/2B-P are samples that were irradiated by an Xe lamp for 70 min; 1B-T/2B-T are samples that were thermally annealed at 180 °C for 150 min; 1A_100% hum/2A_100% hum are samples that were placed in the dark under the 100% relative humidity for 12 h.*”

Meanwhile, for better understanding of this article to the readers, we have re-organized **Figure S5** and **Figure S7** to provide **Figure 4** and **Figure S5**. Accordingly, the titles of **Figure 4** and **Figure S5** are now changed. Please see details in the following “**Miscellaneous**” part.

Miscellaneous:

For better understanding of this article to the readers, we have made the following modifications to the Manuscript and Supporting Information at the same time. (see Manuscript_track.pdf and Supporting information_track.pdf)

1. We have modified **Figure 1** to the following revised figure, which is more intuitive and easy to understand.

■ Previous version:

■ Revised version:

Figure 1. Design strategy in this work. δ and $d_{\pi-\pi}$ denote the interannular angle and common separation for $\pi-\pi$ interactions, respectively.

2. The color of the octahedron in the original version does not match well with those of the central atoms. So, we have modified **Figure 2** to the following revised figure, which is more intuitive and easy to understand.

■ Previous version:

■ Revised version:

Figure 2. Crystal structure of **1**: (a) side view of the 3-D packing structure; b) two π -stacked MQ^+ cations; c) infinitely π -stacked MQ^+ cations between two perovskite-like cyanide-bridged layers (cyano groups are drawn as vertexes of octahedra). Dash lines depict π stacking interactions.

3. We re-organized **Figure S5** and **Figure S7** to highlight the ultrabroad absorption and the high-stability of the radical products. The resulted figures are named as **Figure 4** in the manuscript and **Figure S5** in the supporting information. We think that these new figures are more intuitive and easy to understand. Owing to these changes, the sequence numbers of other figures are changed accordingly.

■ Previous version:

Previous **Figure S5**:

Previous **Figure S7**:

■ **Revised version:**

Revised **Figure 4**:

Figure 4. a) Stability test for the thermo-induced sample **1B-T** monitored through the combination of UV/Visible/NIR spectra (200–2500 nm) and IR spectra (2500–3000 nm). The sample was placed in air in the dark at room temperature for 6 month. The minor difference between **1B-T** and **1B-T@6month** was attributed to systematic errors caused by instrumental instability or shift of the sample holder. b) ESR patterns for **1A**, **1B-P** and **1B-T**. Note: **1A** is as-prepared crystalline sample; **1B-P** is sample that was irradiated by an Xe lamp for 70 min; **1B-T** is sample that was thermally annealed at 180 °C for 150 min.

Revised **Figure S5**:

Figure S5. a) Stability test for the thermo-induced sample **2B-T** monitored using UV/Visible/NIR spectra (200–2500 nm) and IR spectra (2500–3000 nm). The sample was placed in air in the dark at room temperature for 6 month. The minor difference between **2B-T** and **2B-T@6month** was attributed to systematic errors caused by instrumental instability or shift of the sample holder. b) ESR patterns for **2A**, **2B-P** and **2B-T**. Note: **2A** is as-prepared crystalline sample; **2B-P** is sample that was irradiated by an Xe lamp for 70 min; **2B-T** is sample that was thermally annealed at 180 °C for 150 min.

4. To support newly recorded data that are related to the **Comment 2 of Reviewer 2** and **Comment 3 of Reviewer 3**, we have divided the original **Figure S11** into revised **Figure S11** and **Figure S12**.

■ **Previous version:**

Previous **Figure S11**:

a) Sample 1 (described in Figure 4)

b) Sample 2

c) Sample 3

d)

■ **Revised version:**

Revised **Figure S11**:

a) Sample 1 (described in Figure 4)

b) Sample 2

c) Sample 3

Figure S11. Photoswitching behavior for three samples of **1B-T** monitored using lasers with different wavelengths. Power for each wavelength: a) 355 nm, 2.1 W; 600 nm, 5.0 mW; 1500 nm, 23 mW; 2400 nm, 6.0 mW; b) 355 nm, 2.1 W; 600 nm, 12 mW; 1500 nm, 12 mW; 2400 nm, 3.0 mW; c) 355 nm, 2.1 W; 600 nm, 16 mW; 1500 nm, 15 mW; 2400 nm, 3.5 mW.

Revised **Figure S12**:

Figure S12. Laser power-dependent photoresponse behavior of **1B-T** with 1000 nm, 1200 nm and 1500 nm lasers, respectively. Inset: Plot of relative photocurrent gain versus power. As can be seen, photocurrent gain shows a linear correlation to power, which means that a higher power density produces more carriers in a specific optical power range.

5. Due to the above modifications to the figures, we have made corresponding changes to the cited figure numbers in the Manuscript. Relevant changes have been given in the revised Manuscript.

REVIEWERS' COMMENTS:

Reviewer #2 (Remarks to the Author):

The authors give a detailed elucidation. I think this revised version is now acceptable.

Reviewer #3 (Remarks to the Author):

After the revision of the manuscript, the concerning issues have been addressed and therefore I recommend it's suitability for publication.